# Off-pump versus on-pump coronary artery surgery in octogenarians (from the KROK Registry)

Piotr Knapik[1]*, Grzegorz Hirnle[2], Anetta Kowalczuk-Wieteska[2], Michał O. Zembala[2], Szymon Pawlak[2], Tomasz Hrapkowicz[2], Piotr Przybyłowski[3,4], Paweł Nadziakiewicz[1], Daniel Cieśla[5], Bartłomiej Perek[6], Bogusław Kapelak[7], Marek Cisowski[8], Jan Rogowski[9], Edward Pietrzyk[10], Zdzisław Tobota[11], Marian Zembala[2], on behalf of KROK Investigators[¶]

1 Department of Anaesthesiology, Intensive Therapy and Emergency Medicine, Silesian Centre for Heart Diseases, Medical University of Silesia, Zabrze, Poland, 2 Department of Cardiac, Vascular and Endovascular Surgery and Transplantology, Silesian Centre for Heart Diseases, Medical University of Silesia, Zabrze, Poland, 3 Silesian Centre for Heart Diseases, Zabrze, Poland, 4 First Chair of General Surgery, Jagiellonian University, Medical College, Cracow, Poland, 5 Department of Science and New Technologies, Silesian Centre for Heart Diseases, Zabrze, Poland, 6 Department of Cardiac Surgery and Transplantology, Poznan University of Medical Sciences, Poznan, Poland, 7 Department of Cardiovascular Surgery and Transplantology, Jagiellonian University Medical College, John Paul II Hospital, Krakow, Poland, 8 First Department of Cardiac Surgery, American Heart of Poland, Bielsko-Biala, Poland, 9 Department of Cardiac and Vascular Surgery, Medical University of Gdańsk, Gdańsk, Poland, 10 Department of Cardiac Surgery, Świętokrzyskie Centre of Cardiology, Kielce, Poland, 11 Department of Paediatric Cardiothoracic Surgery, Children's Memorial Health Institute, Warsaw, Poland

¶ KROK Investigators list in the Acknowledgments section.
* pknapik@sum.edu.pl

**Data Availability Statement:** All relevant data are within the manuscript and its Supporting Information files.

## Abstract

### Background

According to the medical literature, both on-pump and off-pump coronary artery surgery is safe and effective in octogenarians.

### Objectives

The aim of our study was to examine the epidemiology, in-hospital outcomes and long-term follow-up results in octogenarians undergoing off-pump and on-pump coronary artery surgery utilizing nationwide registry data.

### Methods

All octogenarians ($\geq$ 80 years) enrolled in the Polish National Registry of Cardiac Surgical Procedures (KROK Registry), who underwent isolated coronary surgery between January 2006 and September 2017 were identified. Preoperative data, perioperative complications, hospital mortality and long-term mortality were analyzed. Unadjusted and propensity-matched comparisons were performed between octogenarians undergoing off-pump and on-pump coronary artery bypass surgery.

**Funding:** The authors received no specific funding for this work.

**Competing interests:** The authors have declared that no competing interests exist.

**Abbreviations:** CABG, coronary artery bypass grafting; KROK, Polish National Registry of Cardiac Surgical Operations; MIDCAB, Minimally Invasive Direct Coronary Artery Bypass; OPCAB, Off-pump coronary artery bypass.

## Results

Octogenarians accounted for 4.1% of the total population undergoing coronary artery surgery in Poland during the analyzed period (n = 152,631) and this percentage is increasing. Among 6,006 analyzed patients, 2,744 (45.7%) were operated on-pump and 3,262 (54.3%) were operated off-pump. Propensity-matched analysis revealed that patients operated on-pump were more often reoperated due to postoperative bleeding and their in-hospital mortality was higher (6.6% vs 4.5%, p = 0.006 and 8.7% vs 5.8%, p = 0.001, respectively). Long-term all-cause mortality was lower among patients operated off-pump (p = 0.013).

## Conclusion

On the basis of our findings we suggest that off pump technique should be considered as perfectly acceptable in octogenarians.

## Introduction

In the industrialized world, the average life expectancy continues to increase. Cardiac surgery is confronted with a growing population of octogenarians with coronary artery disease and clear indications for coronary artery surgery [1].

Patients over 80 years old are more prone to increased postoperative morbidity and mortality, because of more frequent comorbid risk factors and frailty [2, 3]. This population requires special care, attention, and treatment. Off-pump coronary artery bypass (OPCAB) grafting gives the surgeon the opportunity to avoid the inherent risks associated with coronary artery bypass surgery (CABG) with cardiopulmonary bypass. These increased risks include hemodilution, global myocardial ischemia, nonpulsatile arterial flow, systemic inflammatory response, and atherosclerotic embolization from aortic manipulation [4].

The debate concerning the superiority of OPCAB over CABG (and vice versa) has continued for many years [4, 5]. Proponents of the OPCAB technique have advocated for its specific use in octogenarians since the beginning of this debate [6] and many studies continue to evaluate the potential benefit [7]. Evidence from meta-analyses indicates that utilizing the off-pump technique significantly reduces stroke, renal failure, ventilation time, atrial fibrillation, transfusion requirements, and postoperative length of stay when compared with conventional CABG using cardiopulmonary bypass [8, 9]. In spite of these promising results, well-designed randomized controlled trials have been unable to consistently demonstrate such benefits [5, 10] and some trials have even cast doubt on the long-term benefit of OPCAB in terms of graft patency [11].

In higher risk octogenarian patients, there is potential for more tangible clinical benefit when cardiopulmonary bypass is avoided [12]. In contrast to this, the most discussed potential benefit of on-pump technique has always been the potential for higher long-term graft patency [11]. But is long-term graft patency really a key issue for octogenarians?

The aim of this retrospective study was to compare the perioperative and long-term results of octogenarians undergoing OPCAB and CABG using data from the nationwide Polish National Registry of Cardiac Surgical Procedures (KROK Registry).

## Methods

### Study design

This analysis is based on data from the Polish National Registry of Cardiac Surgical Operations (KROK Registry), a joint initiative of the Polish Society of Cardiothoracic Surgeons and the Polish Ministry of Health. Details regarding the KROK Registry and the collection of follow-up data have been previously described [13]. Due to the retrospective and anonymous nature of the study, Ethical Committee of the Medical University of Silesia in Katowice waived the need for consent of the patients to participate in the study.

Our study included all octogenarians ($\geq$ 80 years) who underwent isolated coronary artery surgery in Poland between January 2006 and September 2017 (11 years and 9 months). Patients undergoing Minimally Invasive Direct Coronary Artery Bypass (MIDCAB) and patients in whom the type of coronary surgery could not be clearly determined, were excluded. The remaining population of octogenarians undergoing isolated coronary artery surgery was divided into two groups: patients who underwent surgery either on-pump or off-pump (Fig 1). Conversions from OPCAB do CABG were analyzed in the OPCAB group.

To enhance statistical comparison between the groups, the overall population of patients undergoing CABG and OPCAB was then restricted to propensity-matched groups.

The primary outcomes of this study were the in-hospital mortality rate and the incidence of perioperative complications. The secondary outcome was all-cause mortality in a long-term follow-up period.

Each patient was described in terms of baseline demographic data, circulatory function, individual risk factors, general condition directly before the procedure, and procedure-related variables (Table 1).

### Assessment of complications

Patients who developed postoperative complications were identified. Neurological complications were defined to include patients who developed a new neurological deficit in the postoperative period with persistent symptoms still present at the time of the hospital discharge. Respiratory complications included patients who required prolonged mechanical ventilation for more than 24 hours, and/or patients who developed pneumonia in the postoperative period. Renal complications included patients who required any form of renal replacement therapy in the postoperative period. Gastrointestinal complications included patients with gastrointestinal bleeding, pancreatitis, cholecystitis, and/or mesenteric ischemia—with or without a surgical intervention. Sternal, mediastinal or wound infection included all types of surgical site infections of the sternum. Perioperative myocardial infarction was recognized according to the criteria used by the Society of Thoracic Surgeons adult cardiac surgery database. Mechanical circulatory support was broadly defined to include the use any of the available options in this field. ICU readmission was identified if a patient was transferred to the ICU again following a previous discharge from this unit, during the same hospital admission.

### Statistical analysis

Continuous variables were presented as mean and standard deviation, while categorical variables were presented as percentages. Chi-squared, Mann-Whitney U and t-Student tests were used to assess for statistical significance where appropriate. Patients for comparison were matched to achieve the similar preoperative status. The data were matched with the Greedy data matching algorithm using Mahalanobis distance within propensity score calipers. Each caliper radius was set to 0.2*Sigma. Propensity score was calculated using logistic regression.

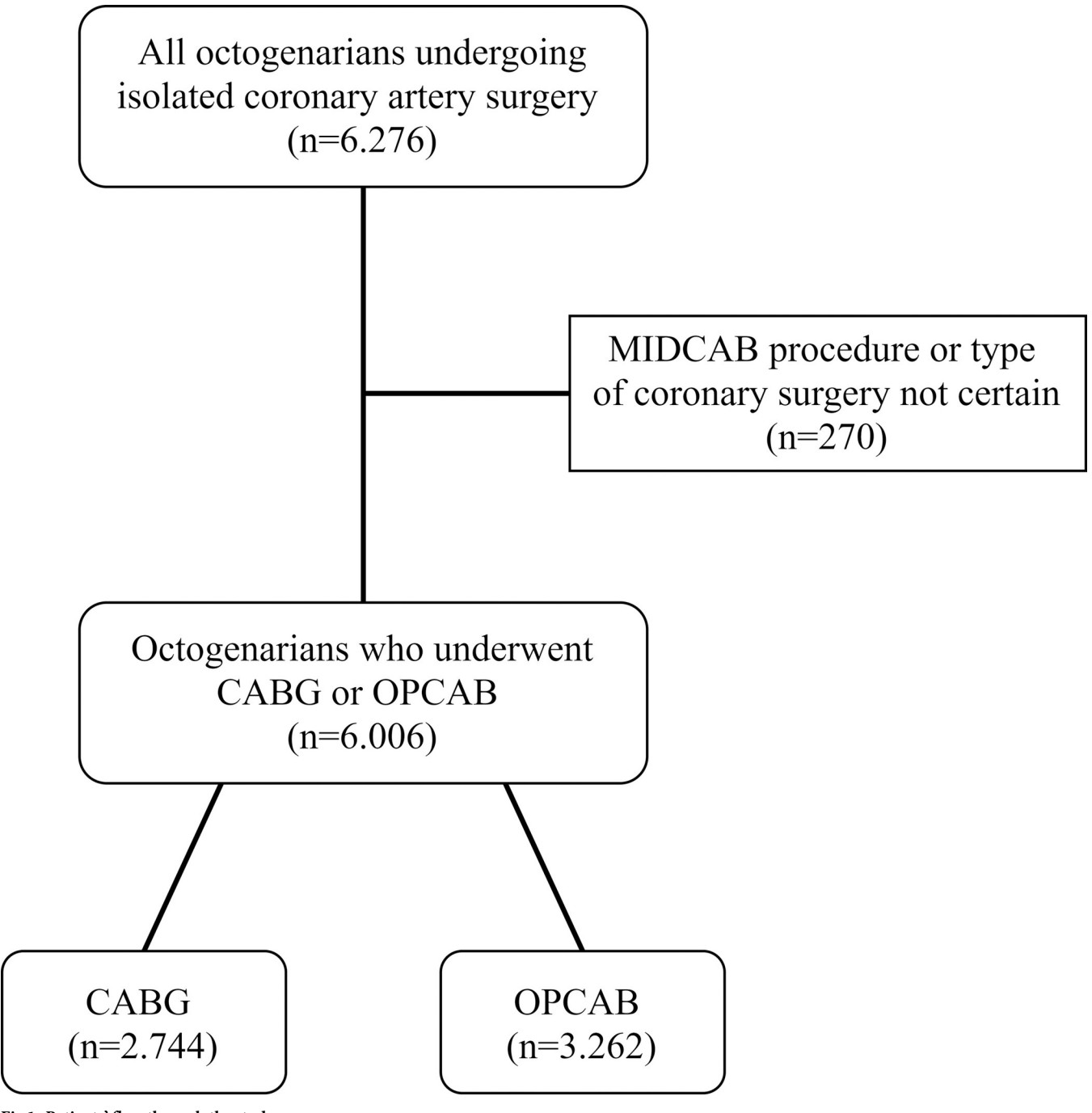

**Fig 1. Patients' flow through the study.**

We used all variables from Table 1. To assess the covariate balance, z-difference coefficients were calculated for each variable before and after matching. The mean value before and after the matching was 0.28 and -0.30, and the variance was 79.31 and 0.32, respectively.

Assessment of long-term follow-up data included analysis of all-cause mortality. The National Health Fund death database was searched for all patients included in this study from the date of their procedure until 30th of September 2017. These data were then

**Table 1. Comparison of preoperative variables in all patients (left) and propensity-matched patients (right).**

| Group of variables | Variable | All patients 80+ | | | | Matched patients 80+ | | | |
| | | CABG | OPCAB | p | z-diff | CABG | OPCAB | p | z-diff |
| | | (n = 2,744) | (n = 3,262) | | | (n = 1,813) | (n = 1,813) | | |
|---|---|---|---|---|---|---|---|---|---|
| **Demographic data** | Age>90 years | 155 (5.6%) | 222 (6.8%) | 0.074 | -1.86 | 112 (6.2%) | 116 (6.4%) | 0.837 | -0.27 |
| | Age 86–90 years | 8 (0.3%) | 16 (0.5%) | 0.311 | -1.24 | 8 (0.4%) | 8 (0.4%) | 0.802 | 0.00 |
| | Female gender | 941 (34.3%) | 1187 (36.4%) | 0.096 | -1.69 | 643 (35.5%) | 636 (35.1%) | 0.835 | 0.24 |
| Circulatory function | CCS class IV | 588 (21.4%) | 645 (19.8%) | 0.121 | 1.58 | 368 (20.3%) | 363 (20.0%) | 0.868 | 0.21 |
| | NYHA class III or IV | 451 (16.4%) | 522 (16.0%) | 0.675 | 0.45 | 282 (15.6%) | 295 (16.3%) | 0.586 | -0.59 |
| | Recent MI<90 days | 908 (33.1%) | 923 (28.3%) | 0.000 | 4.01 | 566 (31.2%) | 589 (32.5%) | 0.433 | -0.82 |
| | Pulmonary hypertension | 5 (0.2%) | 17 (0.5%) | 0.051 | -2.26 | 5 (0.3%) | 4 (0.2%) | 1.000 | 0.33 |
| | LVEF <30% | 90 (3.3%) | 92 (2.8%) | 0.337 | 1.03 | 47 (2.6%) | 52 (2.9%) | 0.684 | -0.51 |
| | Previous PCA/stent | 628 (22.9%) | 597 (18.3%) | 0.000 | 4.37 | 378 (20.8%) | 382 (21.1%) | 0.903 | -0.16 |
| | Persistent or chronic AF | 249 (9.1%) | 335 (10.3%) | 0.130 | -1.57 | 184 (10.1%) | 197 (10.9%) | 0.516 | -0.70 |
| | Left main stem lesion | 1166 (42.5%) | 973 (29.8%) | 0.000 | 10.23 | 677 (37.3%) | 683 (37.7%) | 0.864 | -0.21 |
| | Triple vessel disease | 1876 (68.4%) | 1422 (43.6%) | 0.000 | 19.95 | 1056 (58.2%) | 1090 (60.1%) | 0.265 | -1.15 |
| Individual risk factors | Cigarette smoking | 82 (3.0%) | 130 (4.0%) | 0.044 | -2.11 | 66 (3.6%) | 75 (4.1%) | 0.492 | -0.77 |
| | Hypercholesterolaemia | 1598 (58.2%) | 1771 (54.3%) | 0.002 | 3.07 | 1055 (58.2%) | 1084 (59.8%) | 0.344 | -0.98 |
| | Diabetes mellitus | 879 (32.0%) | 1117 (34.2%) | 0.075 | -1.81 | 655 (36.1%) | 643 (35.5%) | 0.703 | 0.42 |
| | Arterial hypertension | 2438 (88.8%) | 2613 (80.1%) | 0.000 | 9.49 | 1631 (90.0%) | 1631 (90.0%) | 0.956 | 0.00 |
| | BMI>35 kg/m$^2$ | 71 (2.6%) | 78 (2.4%) | 0.686 | 0.49 | 51 (2.8%) | 51 (2.8%) | 0.920 | 0.00 |
| | Renal failure | 379 (13.8%) | 437 (13.4%) | 0.667 | 0.47 | 262 (14.5%) | 274 (15.1%) | 0.607 | -0.56 |
| | COPD | 173 (6.3%) | 209 (6.4%) | 0.913 | -0.16 | 120 (6.6%) | 124 (6.8%) | 0.842 | -0.27 |
| | Past TIA. RIND, stroke | 123 (4.5%) | 123 (3.8%) | 0.186 | 1.38 | 73 (4.0%) | 70 (3.9%) | 0.865 | 0.26 |
| | Past treatment of CAD | 30 (1.1%) | 28 (0.9%) | 0.427 | 0.92 | 20 (1.1%) | 19 (1.0%) | 1.000 | 0.16 |
| | PVD | 352 (12.8%) | 349 (10.7%) | 0.012 | 2.54 | 202 (11.1%) | 210 (11.6%) | 0.714 | -0.42 |
| | Poor mobility | 113 (4.1%) | 175 (5.4%) | 0.028 | -2.28 | 81 (4.5%) | 87 (4.8%) | 0.693 | -0.47 |
| Condition before the procedure | Cardiogenic shock | 101 (3.7%) | 207 (6.3%) | 0.000 | -4.78 | 81 (4.5%) | 69 (3.8%) | 0.359 | 1.00 |
| | Use of IABP | 87 (3.2%) | 55 (1.7%) | 0.000 | 3.68 | 36 (2.0%) | 43 (2.4%) | 0.495 | -0.80 |
| | IV nitrates or heparin. | 479 (17.5%) | 540 (16.6%) | 0.372 | 0.93 | 297 (16.4%) | 315 (17.4%) | 0.451 | -0.80 |
| Procedure-related variables | Previous cardiac surgery | 41 (1.5%) | 39 (1.2%) | 0.372 | 1.00 | 24 (1.3%) | 26 (1.4%) | 0.887 | -0.28 |
| | Non-elective surgery | 1175 (42.8%) | 1438 (44.1%) | 0.339 | -0.98 | 803 (44.3%) | 826 (45.6%) | 0.463 | -0.77 |
| | Complete arterial revascularization | 95 (3.5%) | 897 (27.5%) | 0.000 | -28.08 | 93 (5.1%) | 107 (5.9%) | 0.344 | -1.02 |
| Number of grafts | 1 graft | 85 (3.1%) | 838 (25.7%) | 0.000 | -27.11 | 83 (4.6%) | 93 (5.1%) | 0.487 | -0.77 |
| | 2 grafts | 1046 (38.1%) | 1345 (41.2%) | 0.015 | -2.46 | 898 (49.5%) | 904 (49.9%) | 0.868 | -0.20 |
| | 3 or more grafts | 1613 (58.8%) | 1079 (33.1%) | 0.000 | 20.57 | 832 (45.9%) | 816 (45.0%) | 0.617 | 0.53 |
| Year of surgery | 2006–2009 | 588 (21.4%) | 632 (19.4%) | 0.053 | 1.97 | 341 (18.8%) | 344 (19.0%) | 0.932 | -0.13 |
| | 2010–2013 | 1100 (40.1%) | 1279 (39.2%) | 0.505 | 0.69 | 699 (38.6%) | 711 (39.2%) | 0.708 | -0.41 |
| | 2014–2017 | 1056 (38.5%) | 1351 (41.4%) | 0.022 | -2.31 | 773 (42.6%) | 758 (41.8%) | 0.638 | 0.50 |

Abbreviations: AF–atrial fibrillation, CAD–carotid artery, CCS–Canadian Coronary Score, COPD–chronic obstructive pulmonary disease, IABP–intra-aortic balloon pump, IV–intravenous, LVEF—Left Ventricular Ejection Fraction, MI–myocardial infarction, NYHA–New York Heart Association, PCA–percutaneous coronary angioplasty, PVD–peripheral vascular disease, RIND–reversible ischaemic neurologic deficit, SR–surgical reexploration, TIA–transient ischaemic attack.

analyzed using the Kaplan-Meier method with stratified log-rank testing. For the purpose of the principal analysis, the date of operation was considered the starting point. For the purpose of the additional censored analysis, the day of hospital discharge was considered the starting point. Completness of follow-up data was calculated according to the method described by Wu et. al. [14].

For all analyses, a two-tailed p-value <0.05 was considered statistically significant. The analyses and graphs were performed with the use of Dell Inc. (2016). Dell Statistica (data analysis software system, version 13) and R version 3.6.1 2019 (the R Foundation for Statistical Computing).

## Results

Overall, 152,631 adult patients underwent isolated coronary artery surgery in Poland during the study period of 11 years and 9 months and were included in the KROK Registry. There were 6,276 octogenarians in this population (4.1%). Among 6,276 octogenarians, there were 270 patients who were excluded (4.3%), because they either underwent the MIDCAB procedure or their type of coronary surgery could not be clearly determined. The remaining 6,006 patients were divided into two groups. The on-pump group consisted of 2,744 patients (45.7%) and the off-pump group consisted of 3,262 patients (54.3%). There were 53 conversions from OPCAB to CABG in the OPCAB group. The mean age of this population was 82.0 years (range 80 to 96 years). Completeness of follow-up data according to Clark's C-index was 78.6% and the modified C*-index was 88.8%. Patients' flow through the study has been shown in Fig 1.

### Epidemiology

The overall percentage of octogenarians steadily increased during the study period, from 1.7% in 2006, to 5.9% in 2015, when a plateau was observed. The total number of coronary artery surgery procedures in Poland remained relatively constant throughout these years but recently started to gradually decrease (Fig 2).

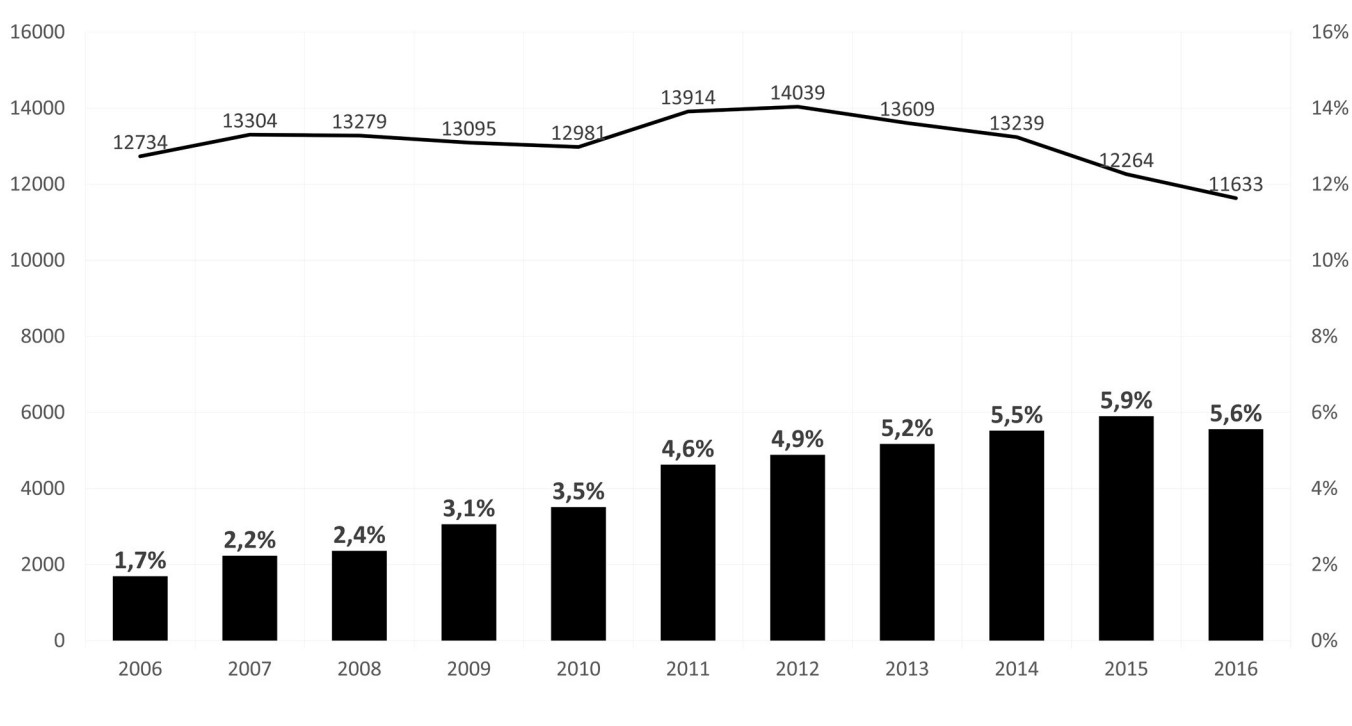

**Fig 2. All coronary artery surgery procedures in Poland over a period of 11 years (2006–2016) and the amount of octogenarians in this population.**

## Comparison of groups

Patients operated on-pump were younger in comparison to those operated off-pump (mean 81.9 ± 2.0 years vs 82.1 ± 2.1 years, p = 0.046). Mean EuroSCORE II values (assessed in all patients since 2012) before matching were found to be significantly higher in patients operated on-pump (4.7 ± 5.8% vs 4.3 ± 5.8%, p<0.001, respectively). Patients operated on-pump generally had more advanced atherosclerosis, as evidenced by a higher incidence of atherosclerosis-related co-morbidities (Table 1).

The on-pump group received a higher mean number of grafts compared to the off-pump group (2.67 ± 0.72 vs 2.16 ± 0.94, p<0.001). Consequently, the mean operative time was significantly longer in patients operated on-pump (3.54 ± 1.23 vs 3.30 ± 1.34 hours, p<0.001). A significantly higher proportion of patients operated on-pump had a procedure time exceeding 3 hours (59.2% vs 51.5%, p<0.001).

The overall time of postoperative ventilation (available in 68.5% of patients in the on-pump and in 63.8% of patients in the off-pump group) was significantly longer for the on-pump group (20.8 ± 47.6 hours vs 15.3 ± 36.8 hours, p<0.001), with a median value of 11.2 vs 9.4 hours, respectively. The percent of patients with postoperative ventilation exceeding 24 hours was significantly higher in patients operated on-pump (11.1% vs 7.5%, p<0.001).

The on-pump group had a higher incidence of major postoperative complications except for neurological complications, renal failure and sternal or wound infection. In addition to this, the in-hospital mortality rate was significantly higher for patients undergoing the on-pump procedure. This result was similar when conversions were analyzed both in the OPCAB group (8.1% vs 5.4%, p<0.001) (Table 2) and in the CABG group (8.5% vs 5.0%, p<0.001).

The overall rate of ICU readmissions was similar (1.1% vs 1.4%, p = 0.397). Mean length of hospital stay was similar in both groups (12.3 ± 8.7 days vs 12.1 ± 8.1 days, p = 0.18), with a similar proportion of patients staying in the hospital longer than two weeks (22.8% vs 23.1%, p = 0.846, respectively).

The mean overall follow-up was 3.6 ± 2.9 years (0–11.7 years) for patients operated on-pump and 3.6 ± 2.8 years (0–11.7 years) for patients operated off-pump. The overall total mortality rate for these groups was 37.2% vs 35.6%, respectively (p = 0.185). Kaplan-Meier estimate of mortality in subsequent time intervals in the first year of the follow-up is presented on a left side of Table 3. Kaplan–Meier follow-up curves for all-cause mortality are presented on a left side of Fig 4. The principal analysis (starting at the day of operation) indicated that long-term

**Table 2. Comparison of postoperative complications in all patients (left) and in propensity-matched patients (right).**

| Postoperative course | All patients | | | Matched patients | | |
|---|---|---|---|---|---|---|
| | CABG | OPCAB | p | CABG | OPCAB | p |
| | (n = 2,744) | (n = 3,262) | | (n = 1,813) | (n = 1,813) | |
| Conversion | 0 (0%) | 53 (1.6%) | - | 0 (0%) | 41 (1.1%) | - |
| Neurological complications | 68 (2.5%) | 60 (1.8%) | 0.106 | 46 (2.5%) | 38 (2.1%) | 0.440 |
| Respiratory complications | 205 (7.5%) | 158 (4.8%) | 0.000 | 134 (7.4%) | 102 (5.6%) | 0.037 |
| Gastrointestinal complications | 46 (1.7%) | 35 (1.1%) | 0.056 | 33 (1.8%) | 25 (1.4%) | 0.354 |
| Renal complications | 119 (4.3%) | 141 (4.3%) | 0.971 | 74 (4.1%) | 90 (5.0%) | 0.231 |
| Sternal, mediastinal or wound infection | 59 (2.2%) | 63 (1.9%) | 0.612 | 37 (2.0%) | 35 (1.9%) | 0.905 |
| Perioperative myocardial infarction | 41 (1.5%) | 27 (0.8%) | 0.021 | 25 (1.4%) | 17 (0.9%) | 0.277 |
| Mechanical circulatory support | 122 (4.4%) | 109 (3.3%) | 0.032 | 78 (4.3%) | 70 (3.9%) | 0.557 |
| ICU readmission | 31 (1.1%) | 46 (1.4%) | 0.397 | 24 (1.3%) | 30 (1.7%) | 0.493 |
| Reoperation due to bleeding | 184 (6.7%) | 138 (4.2%) | <0.001 | 120 (6.6%) | 81 (4.5%) | 0.006 |
| In-hospital death | 221 (8.1%) | 177 (5.4%) | <0.001 | 157 (8.7%) | 106 (5.8%) | 0.001 |

**Table 3. Estimated mortality in subsequent time intervals following hospital discharge in the first year of the follow-up period in all patients (left) and propensity-matched patients (right).**

| | All patients | | | Matched patients | | |
|---|---|---|---|---|---|---|
| | CABG | OPCAB | p | CABG | OPCAB | p |
| | (n = 2,744) | (n = 3,262) | | (n = 1,813) | (n = 1,813) | |
| In-hospital mortality | 221 (8.1%) | 177 (5.4%) | <0.001 | 157 (8.7%) | 106 (5.8%) | **0.001** |
| Discharge (days) | | | | | | |
| 30 | 54 (2.2%) | 64 (2.1%) | 0.928 | 31 (1.9%) | 31 (1.8%) | 0.997 |
| 31–60 | 64 (2.6%) | 55 (1.8%) | 0.060 | 42 (2.6%) | 31 (1.9%) | 0.183 |
| 61–90 | 28 (1.2%) | 28 (1.0%) | 0.506 | 21 (1.4%) | 13 (0.8%) | 0.184 |
| 91–180 | 28 (1.2%) | 62 (2.2%) | **0.012** | 21 (1.4%) | 31 (2.0%) | 0.282 |
| 181–270 | 33 (1.5%) | 43 (1.6%) | 0.871 | 20 (1.4%) | 11 (0.7%) | 0.113 |
| 270–365 | 25 (1.2%) | 38 (1.5%) | 0.459 | 18 (1.3%) | 20 (1.4%) | 0.990 |

all-cause mortality was similar in both groups (p = 0.157). Additional censored analysis (starting at the day of hospital discharge) indicated that long-term all-cause mortality was also similar in both groups (p = 0.232) (see S1A Fig to this manuscript).

## Comparisons of propensity-matched groups

The mean age of propensity-matched groups was similar for on-pump and off-pump coronary artery surgery (82.0 ± 2.1 years vs 82.0 ± 2.1 years, p = 0.30). Mean EuroSCORE II values were similar in both groups (4.5 ± 5.4% vs 4.3 ± 5.6%, p = 0.103, respectively). Following the propensity-matching procedure, all preoperative differences in patients' baseline demographics became non-significant (Table 1).

Following propensity-score matching, patients operated on-pump received a similar mean number of grafts compared to the off-pump group (2.48 ± 0.71 grafts vs 2.51 ± 0.78 grafts, p = 0.930). The mean operative time following matching was similar in both groups (3.49 ± 1.24 hours vs 3.43 ± 1.29 hours, p = 0.066), as was the proportion of patients whose procedural time exceeded 3 hours (58.0% vs 56.0%, p = 0.248).

The total time of postoperative ventilation (available in 71.1% of patients in the on-pump group and in 67.3% of patients in the off-pump group) was longer among patients operated on-pump (20.3 ± 49.0 hours vs 15.2 ± 38.5 hours, p<0.001), with a median value of 11.7 vs 9.3 hours, respectively. The percentage of patients with postoperative ventilation exceeding 24 hours was significantly higher in the on-pump cohort (10.9% vs 8.1%, p = 0.019). A graphical comparison of propensity-matched patients remaining ventilated in consecutive postoperative hours has been additionally expressed in the form of a Kaplan-Meier curve (Fig 3).

Following propensity-matching, the incidence of postoperative respiratory complications and reoperation due to bleeding was higher in the on-pump group. In-hospital mortality also remained significantly higher in the CABG group. This result was similar when conversions were analyzed both in the OPCAB group (8.7% vs 5.8%, p = 0.001) (Table 2) and in the CABG group (9.3% vs 5.1%, p<0.001). The average hospital length of stay and the proportion of patients with a hospital stay longer than two weeks was similar in a both groups (12.3 ± 8.8 days vs 12.3 ± 8.2 days, p = 0.74 and 22.5% vs 24.1%, p = 0.270, respectively).

The mean follow-up period was 3.4 ± 2.8 years (range 0–11.7 years) for patients operated on-pump and 3.6 ± 2.7 years (range 0–11.7 years for patients operated off-pump). The overall mortality rate for these groups was 36.1% vs 33.9%, respectively (p = 0.175). Kaplan-Meier estimate of mortality in subsequent time intervals in the first year of the follow-up is presented on a right side of Table 3. Kaplan–Meier follow-up curves for all-cause mortality are presented on

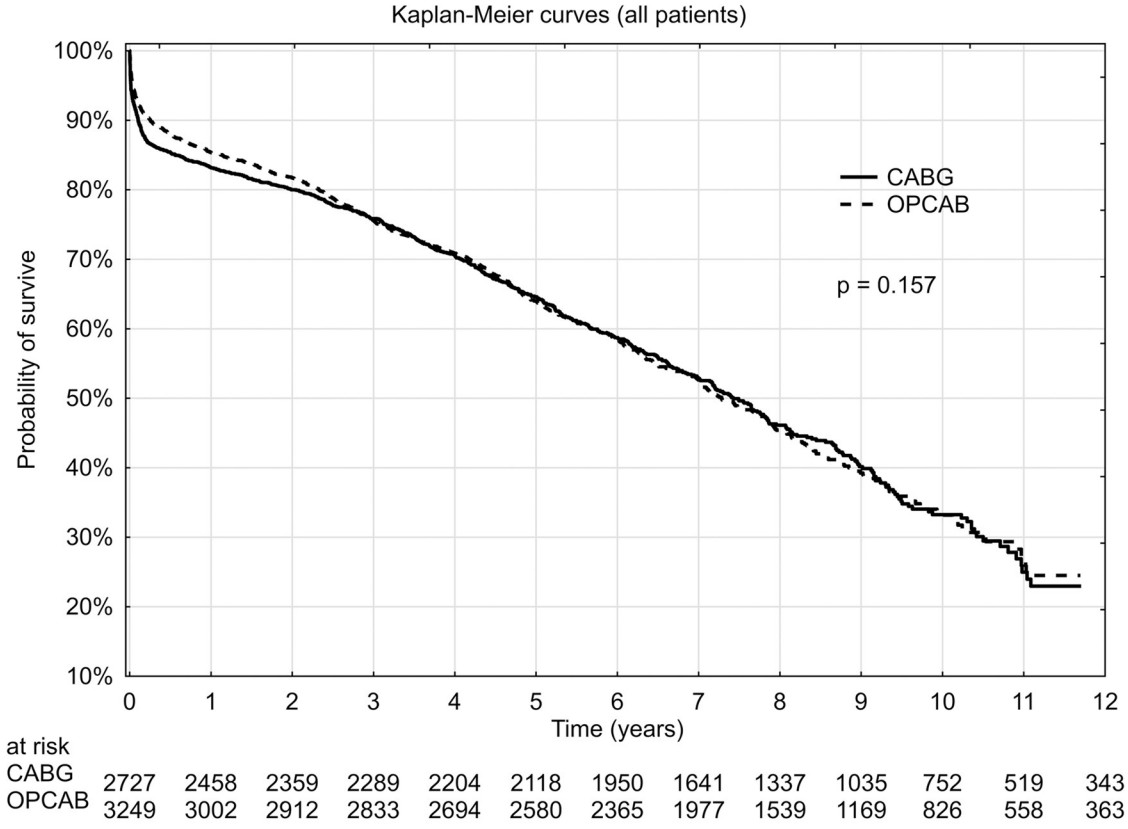

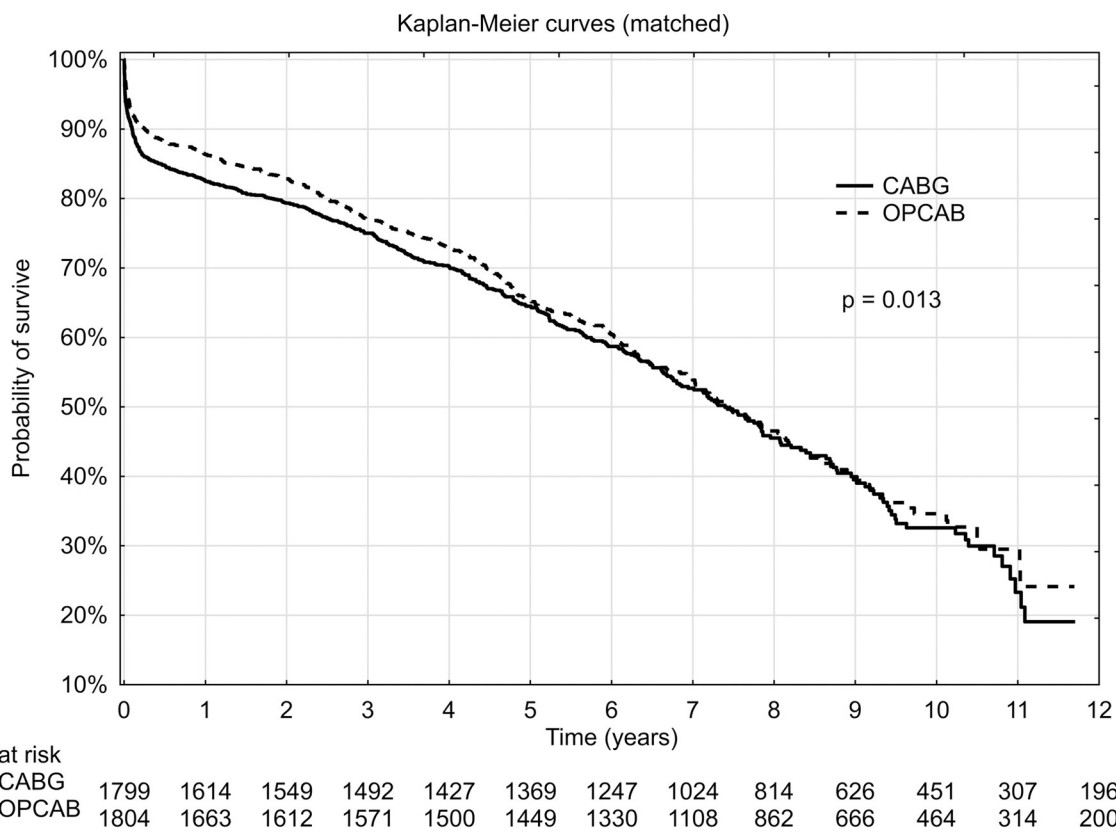

**Fig 4.** Probability of survival in octogenarians undergoing CABG and OPCAB surviving hospital stay: (A)–all patients, (B)–propensity-matched patients.

a right side of Fig 4. The principal analysis (starting at the day of operation) indicated that long-term all-cause mortality was lower in the OPCAB group (p = 0.013). Additional censored analysis (starting at the day of hospital discharge) indicated however, that long-term all-cause mortality was similar in both groups (p = 0.362) (see S1B Fig to this manuscript)

## Discussion

Our findings suggest that octogenarians constitute a high-risk population and the off-pump procedure appears superior among patients scheduled for isolated coronary artery bypass surgery for these patients in terms of in-hospital results. Utilization of the on-pump technique resulted in significantly higher in-hospital mortality (8.7% vs 5.8%, p<0.001) and higher all-cause mortality in the long-term follow-up period (p = 0.013). The increased in-hospital mortality was likely due to the higher incidence of postoperative complications (mainly reoperation due to bleeding) in the on-pump group. Similar conclusions were drawn by Hulde et al.

Kaplan-Meier curves (matched)

at risk

| | | | | | | | | | | | | |
|---|---|---|---|---|---|---|---|---|---|---|---|---|
| CABG | 1238 | 1228 | 1200 | 1071 | 874 | 685 | 554 | 446 | 365 | 290 | 209 | 161 | 142 |
| OPCAB | 1175 | 1156 | 1055 | 903 | 690 | 512 | 410 | 322 | 252 | 207 | 159 | 115 | 99 |

**Fig 3. Kaplan-Meier curves for octogenarians remaining ventilated following CABG and OPCAB in a propensity-matched patients.**

who observed that the duration of mechanical ventilation, intensive care unit stay, the risk of stroke, in-hospital mortality and 30-day mortality were significantly lower in the off-pump group than in the on-pump group [15].

Comparative findings reported in the literature are contradictory. One recent retrospective study, analyzing 134,117 discharge records from 797 US hospitals, found that in-hospital mortality did not differ between octogenarians who underwent CABG and OPCAB (5.5% vs. 5.2%, p = 0.3) [16]. At the same time, a recent systematic review based on 16 retrospective studies (performed in 27,623 octogenarians overall) found that the OPCAB technique was associated with significantly lower in-hospital mortality [9]. The other large study, a US register analysis by Chikwe et al., reported higher mortality over 10-years of follow-up in the off-pump versus on-pump group [17].

With respect to prospective studies, Diegeler et al. recently published the long-term results of the European GOPCABE trial on 2,539 patients aged 75+ who underwent coronary surgery in Germany between 2008 and 2011 [11]. The study found no significant difference between CABG and OPCAB with respect to the 5-year survival rates and the combined outcome of death, myocardial infarction, and repeat revascularization [11]. An earlier interim report from the same population confirmed similar short term results in these two groups [5]. In the prospective, European DOORS trial, Houlind et al. also found that OPCAB technique was non-inferior to CABG in a group of patients aged 70+ [1], despite lower graft patency [10].

In Europe, there are only a few retrospective studies comparing the results of CABG and OPCAB in octogenarians with an appropriate sample size [9]. Based on our review of the literature, our study currently represents the largest (and most contemporary) analysis of octogenarians undergoing coronary surgery in Europe. It is also worth emphasizing that our study comes from a geographical area with only few available data in this field so far, where a rapid increase in the number of cardiac surgical procedures is occurring [13].

According to the KROK database, the overall proportion of octogenarians undergoing coronary artery surgery has been steadily increasing from 1.7% of all coronary artery procedures in 2006 to 5.9% in 2015. These findings are consistent with the trends reported in other countries. For example, octogenarians comprised 8.2% of a contemporary US population scheduled for isolated coronary surgery [7]. The recently observed plateau with further decline may be related to the rapid advances in non-surgical therapies, including implantation of drug-eluting stents and the implementation of newer anticoagulant and antiplatelet medications in combination with aggressive lipid-lowering therapy. These advances clearly improve the results of less invasive treatment of patients with coronary artery disease [9, 18]. Among those octogenarians who still undergo surgery, the percentage of on-pump and off-pump procedures is currently very similar.

The decision regarding surgical technique is always the surgeon's individual decision. In our study, Polish octogenarians operated on-pump were found to have a higher rate of advanced atherosclerosis. Opposite trends might be found in a retrospective analysis performed by Cavallaro et al. based on the largest publicly available database of inpatient hospital care in the US, where data of 187,366 patients undergoing CABG and 69,779 patients undergoing OPCAB were compared (all ages, octogenarians included) [19]. Patients undergoing OPCAB differed preoperatively from those scheduled for on-pump surgery. The strongest independent predictors of off-pump use were the presence of aortic atherosclerosis, liver disease and renal failure; whereas, patients with diabetes, previous myocardial infarction, previous cardiac surgery or percutaneous coronary intervention were more likely to undergo on-pump surgery [19]. Because the study did not assess these factors by age, it is impossible to know whether a similar trend was present for octogenarian patients. Moreover, the Cavallaro

study was based on patients who underwent surgery from 2005–2010, and our data come from the years 2006–2017 (with the majority of octogenarians scheduled for surgery after 2010).

The literature regarding the demographic characteristics of patients who undergo off-pump and on-pump surgery is inconsistent. Benedetto et al. observed a higher percentage of patients with peripheral vascular disease, chronic pulmonary disease and congestive heart failure among 137,117 octogenarians who underwent OPCAB in comparison to patients undergoing CABG [16]. These findings were not replicated in a 2015 systematic review of 9,744 CABG and 8,566 OPCAB patients [12], or in the preoperative data previously presented by LaPar et al. [20]. Significant differences have been also observed between countries [21]. Therefore, an overall, cohesive assessment of the differences between patients operated with the use of CABG or OPCAB technique is not possible given the current state of the literature.

Both unadjusted and propensity-matched analyses revealed that on-pump myocardial revascularisation in Polish octogenarians was associated with more frequent postoperative complications, with the resulting increased in-hospital mortality. Patients operated on-pump were more frequently reoperated due to bleeding–this difference was particularly striking and independent of propensity matching. Respiratory complications such as prolonged postoperative ventilation (over 24h) also appeared more frequently in the CABG group. Both complications are well-known to affect in-hospital mortality [13].

The association between reoperation due to bleeding and overall survival has previously been described by our group [13]. Additionally, the increased frequency of respiratory complications among our patients operated on-pump is likely related to their prolonged times of postoperative ventilation. In a recent prospective randomized trial, patients operated on-pump also had a ventilation time twice as long as their counterparts operated off-pump [22].

In a previously mentioned Nationwide Inpatient Sample study, the authors reported a higher rate of stroke among octogenarians operated on-pump [16]. The same difference was noted in a recent systematic review concentrating on stroke rates among octogenarians undergoing coronary surgery [12]. In our study, it should be noted that the difference in the incidence of neurological complications was significant in an unadjusted comparison, but there was no statistically significant difference after propensity-matching.

Other factors analyzed in our study, such as renal complications, wound infections, and ICU readmissions did not reach statistical significance. In terms of renal failure, our results underscore previous studies based on large databases and meta-analyses, which also reported no statistical difference in the rate of renal replacement therapy among octogenarians in these two groups [16, 23].

Our follow-up results contrast with the conclusions of both the GOPCABE and DOORS trials, where long-term survival was similar also when taking in-hospital death into account [10, 11]. It should be noted however, that patients included in both these trials were younger (75 + and 70+, respectively). These studies were also not based on registry data.

In our study, octogenarians operated on-pump received a higher mean number of grafts. This difference clearly influenced the mean operative time as well as the proportion of patients with a procedural time exceeding 3 hours. The advantage of CABG regarding the number of grafts is supported by nearly every study comparing CABG and OPCAB; however, the ROOBY trial indicated that the number of grafts initially planned per patient was the same in both groups [24]. The CORONARY trial concluded that incomplete revascularisation was similar in both groups (10% for on-pump vs. 11.2% for off-pump; P = 0.05) [25]. In our study, the OPCAB group received a lower number of grafts with a similar long-term survival (among survivors of the hospital stay).

Interestingly, we observed that octogenarians scheduled for off-pump surgery were more likely to be diagnosed with cardiogenic shock or to be on pharmacological inotropic support,

but the pre-procedural intra-aortic balloon pump was more frequently inserted in the on-pump group. There was however, a higher incidence of significant left main stem lesion in the CABG group and preoperative implantation of an intra-aortic balloon pump in such patients is a standard of care in many Polish centers [26, 27].

Our findings based on data from the KROK registry represent the real-life results of coronary surgery in Polish octogenarians, but it has long been known that there is a considerably lower risk of in-hospital mortality in these patients when they are operated off-pump [18]. Moreover, it is also worth mentioning, that the advantage of OPCAB in Polish octogenarians has been just confirmed in a subgroup of patients with specific co-morbidity [28]. Kowalczuk et al. [28] noticed, that among patients with significant left main stenosis, in-hospital mortality was higher only when the procedure was performed on-pump. This difference proved to be significant despite the fact, that sample size in this study was significantly lower, only patients with records indicating preoperative significant left main stenosis were analyzed and the analyzed period was shorter by almost two years [28].

Such strong scientific evidence should lead to careful evaluation of the optimal method of revascularization of octogenarians in Poland. This is also a situation when locally obtained, retrospective data are of great practical importance.

Our study has some important limitations. The study was observational and therefore not randomized and prone to bias. Surgeons' rationale for selection of either the CABG or OPCAB technique in various centers is unknown. It is also important to note that the source of our data is heterogenous. Some of our data come from centers where beating heart surgery is widely used, and therefore the OPCAB technique might be more successful due to a learning curve effect. Because our study utilized registry data, we were strictly limited to the data available in KROK Registry [13]. Finally, our follow-up analysis was limited to all-cause mortality, without the necessary detail to assess whether an octogenarian's death was secondary to the coronary surgery or an unrelated cause.

Based on the results of this study, we suggest that the OPCAB technique should be considered as perfectly acceptable in octogenarians when performed by surgeons experienced in the technique. In these high-risk patients, this technique appears to offer a lower perioperative mortality rate, a lower rate of major perioperative adverse events and better long-term survival.

## Supporting information

**S1 File.**
(XLS)

**S2 File.**
(XLSX)

**S1 Fig.**
(JPG)

## Acknowledgments

The authors wish to thank all KROK Investigators for providing data for this analysis. The authors also wish to thank Jolanta Cieśla for editorial help in preparing the manuscript and Patryk Korecki for statistical support.

KROK Investigators: Lech Anisimowicz, Andrzej Biederman, Dariusz Borkowski, Mirosław Brykczyński, Paweł Bugajski, Paweł Cholewiński, Romuald Cichoń, Marek Cisowski, Marek Deja, Antoni Dziatkowiak, Leszek A. Gryszko, Tadeusz Gburek, Ireneusz Haponiuk, Piotr Hendzel, Tomasz Hirnle, Stanisław Jabłonka, Krzysztof Jarmoszewicz, Marek Jasiński, Ryszard

Jaszewski, Marek Jemielity, Ryszard Kalawski, Bogusław Kapelak, Jacek Kaperczak, Maciej A. Karolczak, Michał Krejca, Wojciech Kustrzycki, Mariusz Kuśmierczyk, Paweł Kwinecki, Bohdan Maruszewski, Maurycy Missima, Jacek J. Moll, Wojciech Ogorzeja, Jacek Pająk, Wojciech Pawliszak, Edward Pietrzyk, Grzegorz Religa, Jan Rogowski, Jacek Różański, Jerzy Sadowski, Girish Sharma, Janusz Skalski, Jacek Skiba, Janusz Stążka, Piotr Stępiński, Kazimierz Suwalski, Piotr Suwalski, Zdzisław Tobota, Łukasz Tułecki, Kazimierz Widenka, Michał Wojtalik, Stanisław Woś, Marian Zembala and Piotr Żelazny.

## Author Contributions

**Conceptualization:** Piotr Knapik, Grzegorz Hirnle, Marian Zembala.

**Data curation:** Daniel Cieśla.

**Formal analysis:** Piotr Knapik, Daniel Cieśla, Zdzisław Tobota.

**Investigation:** Bartłomiej Perek, Bogusław Kapelak, Marek Cisowski, Jan Rogowski, Edward Pietrzyk, Zdzisław Tobota.

**Methodology:** Piotr Knapik, Daniel Cieśla.

**Project administration:** Piotr Knapik.

**Resources:** Zdzisław Tobota.

**Software:** Daniel Cieśla.

**Supervision:** Piotr Knapik.

**Writing – original draft:** Piotr Knapik, Grzegorz Hirnle, Anetta Kowalczuk-Wieteska, Michał O.Zembala, Szymon Pawlak, Tomasz Hrapkowicz, Piotr Przybyłowski, Paweł Nadziakiewicz.

**Writing – review & editing:** Grzegorz Hirnle, Anetta Kowalczuk-Wieteska, Michał O.Zembala, Szymon Pawlak, Tomasz Hrapkowicz, Piotr Przybyłowski, Paweł Nadziakiewicz.

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
