## [Decision Letter · Decision Letter 0]

27 May 2020

PONE-D-20-11113

Off-pump versus on-pump coronary artery surgery in octogenarians (from the KROK Registry)

PLOS ONE

Dear Dr. Knapik,

Thank you for submitting your manuscript to PLOS ONE. After careful consideration, we feel that it has merit but does not fully meet PLOS ONE’s publication criteria as it currently stands. Therefore, we invite you to submit a revised version of the manuscript that addresses the points raised during the review process.

We look forward to receiving your revised manuscript.

Kind regards,

Mariusz Kowalewski

Academic Editor

PLOS ONE

2. In the ethics statement in the manuscript and in the online submission form, please provide additional information about the patient records used in your retrospective study. Specifically, please ensure that you have discussed whether all data were fully anonymized before you accessed them and/or whether the IRB or ethics committee waived the requirement for informed consent. If patients provided informed written consent to have data from their medical records used in research, please include this information.

3. Please note that PLOS journals require authors to make all data underlying the findings described in their manuscript fully available without restriction at the time of publication. When specific legal or ethical requirements prohibit public sharing of a dataset, authors must indicate how researchers may obtain access to the data. Therefore, please update your Data Availability statement to indicate how other researchers may gain access to the KROK Registry that was analysed in the manuscript. For more information, please see: https://journals.plos.org/plosone/s/data-availability

4. We note that there are several previous publications that analyse KROK registry data to draw conclusions about patient outcomes after coronary artery surgery; however, not all of these publications are cited and discussed in the current submission. In your revision, please ensure that all relevant preceding studies are discussed to provide appropriate context for this work, and provide details as to how the current manuscript advances on previous work.

5. One of the noted authors is a group or consortium [KROK Investigators]. In addition to naming the author group, please list the individual authors and affiliations within this group in the acknowledgments section of your manuscript. Please also indicate clearly a lead author for this group along with a contact email address.

Reviewers' comments:

Reviewer's Responses to Questions

**Comments to the Author**

1. Is the manuscript technically sound, and do the data support the conclusions?

Reviewer #1: Partly

Reviewer #2: Yes

Reviewer #3: Partly

Reviewer #4: Partly

2. Has the statistical analysis been performed appropriately and rigorously? 

Reviewer #1: I Don't Know

Reviewer #2: Yes

Reviewer #3: I Don't Know

Reviewer #4: N/A

3. Have the authors made all data underlying the findings in their manuscript fully available?

Reviewer #1: Yes

Reviewer #2: Yes

Reviewer #3: Yes

Reviewer #4: No

4. Is the manuscript presented in an intelligible fashion and written in standard English?

Reviewer #1: Yes

Reviewer #2: Yes

Reviewer #3: Yes

Reviewer #4: Yes

5. Review Comments to the Author

Reviewer #1: The authors compared On-pump vs. off Pump CABG in octgenarians of the Polish cardiac suregery registry. They found that on-Pump was associated with more perioperative complications and mortality but similar long term outcomes.

Comments

This is a well written manuscript, which is easy to comprehend. It addresses a classic but still relevant topic in cardiac surgery, i.e. the comparison of off-pump vs. on-pump bypass surgery. The data are valuable and important. However, the issue of conversions from off-pump to on-pump is not addressed. if this major confounder can be excluded, there remain only minor concerns.

Specific comments

1. One pf the big problems in the comparison of Off-Pump to On Pump is the method of data comparison. In randommized trials with an intention to treat analysis, differences are often not visible, while in retrospective analyses, conversnions from Off- to On-Pump may affect the results. It is not clear in this dataset, how this confounder was handeled. Does the database record conversions? Were they excluded? How many were there? Is the higher mortality in th on-Pump group possibly due to those converted patients ? This needs to be clarified and discussed in detail.

2. The authors state, that it is more difficult to decide for surgery in the elderly today than it was previously. The argument is not convincing, when life expectancy is increaseing. Their figures even illustrate average life expectancy of this patient population in Poland which can easily be used to make recomendations for the only treatment of coronary artery disease with a prognostic impact demonstrated in prospective randomimzed trials.

Reviewer #2: The present study investigated the effect of off-pump versus on-pump isolated CABG surgery on short-term and long-term clinical outcome in the entire cohort of Polish octogenarians who underwent cardiac surgery between 2006 and 2017. It is concluded that the off-pump technique is associated with lower in-hospital mortality than the on-pump technique, whereas long-term mortality is similar between study groups.

General comments

This is a large register analysis of Polish octogenarians. Since there is still a controversial debate regarding long-term clinical outcomes in patients undergoing off-pump or on-pump isolated CABG surgery, data are timely. However, there are issues that have to be addressed:

• Statistical analysis of unadjusted data is subject to unexplained confounding. Therefore, comparison of unadjusted data regarding clinical outcomes are irrelevant. Of scientific interest regarding the effect of off-pump versus on-pump on clinical outcome are only the PS-matched data of the manuscript. Therefore, they should primarily present the PS-matched data both in the Results section as well as in the Discussion section. The manuscript should be revised accordingly. The entire study cohort can be used, however, to perform sensitivity analyses by presenting multivariable-adjusted data (see Puehler et al. Thorac Cardiovasc Surg. 2019 Feb 9. doi: 10.1055/s-0039-1677835. [Epub ahead of print].

• In the Methods section, they should clearly define primary and secondary endpoints. This is necessary because otherwise statistical methods are needed to consider the problem of multiple testing.

• The method of PS-matching should be clearly described in the Statistics section. Moreover, the baseline characteristics used for PS-matching should be listed. Usually, standardized differences rather than p-values are used to compare baseline parameters between the two groups of the entire study cohort and the PS-matched groups. They should revise Table 1 accordingly.

• Clinical outcomes may be influenced by year of surgery, surgeon’s experience, and number of diseased vessels. It is important to consider these parameters in the PS-matched cohort. If some of these data are not available, this should be mentioned as limitation in the Discussion section.

• Several perioperative data, which may substantially influence long-term clinical outcome, are missing and should be presented by study group. These parameters include the revascularization ratio, the percentage of complete arterial revascularizations, and the ratio of posterior to inferior anastomoses.

• Although not restricted to octogenarians, they should refer to two PS-matched large studies: One study supports the early results of the present study regarding beneficial effects in the off-pump versus the on-pump group on clinical outcome (Hulde et al. Interact Cardiovasc Thorac Surg. 2020;30:538-540. The other large study, a US register analysis, reported higher mortality over 10-years of follow-up in the off-pump versus on-pump group (Chikwe et al. J Am Coll Cardiol. 2018;72:1478-1486.). Potential causes of the inconsistent results between the present data and the US data should be discussed more deeply.

Specific comments

• A flowchart of included and excluded patients should be presented.

• Completeness of follow-data should be calculated and presented in the manuscript (see Wu et al. Ann Thorac Surg 2008;85:1155-7).

• Change the titles of figures 3a+3b into ‘Probability of Survival’.

• Since they used the term ‘OPCAB’, they should also use the term ‘ONCAB’ instead of ‘CABG’ throughout the manuscript.

Reviewer #3: This is a retrospective review by Knapik and colleagues of isolated on- and off-pump CABG in octogenarians in Poland as obtained from the Polish National Registry of Cardiac Surgical Procedures (KROK). The primary aims of the study were to evaluate 1. perioperative (in -hospital mortality and complications) and 2. long term outcomes (all cause mortality).

Major Comments:

1. Why use the date of discharge as the starting point for survival; this is a somewhat odd choice? Isn't the day of operation more standard and more representative of survival from an operation? Were perioperative in-hospital deaths eliminated from the survival curves by using the day of discharge as the starting point? what is a patient was discharged 3 months after the operation, the day after discharge is survival day #1?

2. A statement that informed consent was waived by whatever appropriate review board should be included in the paper (if not obtained then rationale for why not should be included but hard to justify)

3. On line 165, you report the mean age was only 82 which means that the vast majority of patients with <85 years old. In Table 1, you report only 39 patients were over 90; can you include additional information about the number of patients over 85 and less than 90? This will help surgeons when applying data to their practices about what "octogenarians" you are operating on.

4. Can you provide any information as to surgeon performance of both procedures? For instance, did surgeons performing OPCAB represent only a small percentage of total surgeons performing most of the operations or were OPCAB's performed by a wide variety of surgeons in a small percentage of patients? It would be important to know if the outcomes could be generalized to essentially any surgeon even if less experienced. Near the end of the discussion, you mention that surgeons' rationale for procedure selection is unknown and also there were centers where OPCAB was widely used (and presumably some where it was rarely used) Can you provide any information as to variation by institution? by surgeon?

5. Although a Kaplan Meier curve can be used to depict the time on ventilator following surgery, the inclusion of 2 figures just to show this before and after propensity matching seems excessive; please remove and instead pick one or two additional individual time points to demonstrate the difference (you talk about the 24 hr time point already in the text). Also, please comment on the "significance" of the difference (13.0% versus 8.2% after 24 hrs); was there higher ventilator associated pneumonias in the patients remaining ventilated. Was there still a significant difference after 48 hrs? What about after 3 or more days?

6. The statement on lines 353-355 "This is a situation when locally obtained, retrospective data should be treated with the same attention as the results of a perfectly designed prospective, randomized trials." is not consistent with known biases that occur with patient selection in retrospective trials which cannot be consistently corrected with "propensity matching" This comment should be removed as "perfectly planned prospective randomized trials remain the gold standard.

7. Your conclusion paragraph states a little too strongly that OPCAB should be the method of choice in octogenarians and this statement and the data may not apply to certain subsets of patient where on pump CABG would be a better choice. This statement should be softened to something like "suggest that the OPCAB technique should be considered as perfectly acceptable in octogenarians when performed by surgeons experienced in the technique."

Minor Comments:

1. Table 2 title: please correct spelling of "patiets" to patients

2. On lines 223-24 the statement ""the incidence of the some postoperative complications was still higher in the on-pump group." does not make sense; please correct

Reviewer #4: PONE-D-20-11113: statistical review

SUMMARY. This is a retrospective study that investigates in-hospital and long-term mortality among octogenarians undergoing off-pump and on-pump coronary artery bypass surgery. The core statistical analysis is based on the comparison of survival curves under off-pump or on-pomp treatments. Although the results seem sound and the material is well organized, the paper lacks details (major issue 1) and the data are not provided (major issue 2). These issues complicate not only the interpretation of the results but also their reproducibility. I also list a couple of specific points that should be addressed.

MAJOR ISSUES

1. Propensity score matching. The paper does not provide enough details about the methods used, making it impossible to reproduce the results. Please provide these details and especially clarify (1) whether you used a logistic regression model to estimate the propensity score (in this case, please provide the estimates, perhaps as supplementary material) and (2) the covariates that have been used (did you consider any variable selection method?).

2. Data availability. Although the authors declare that the data are available without restrictions and that they are within the manuscript and its Supporting Information files, data are not attached. Data should be provided as a supplementary information file along with the metadata needed to process the file.

SPECIFIC POINTS

1. Line 152 “Multivariate analysis of long-term results was performed with the use of the Cox-proportional hazard model.” I can’t see the results of this analysis in the paper. Please clarify.

2. Line 150 “These data were then analyzed using the Kaplan-Meier method with log-rank testing.”. What kind of log-rank test? While the standard log-rank test is frequently used for testing the equality of survival curves in propensity score matched samples, such an approach is inappropriate, because it requires that the samples be independent of one another. Instead, the stratified log-rank test can be used to compare the equality of the survival curves in matched samples.

6. PLOS authors have the option to publish the peer review history of their article (what does this mean?). If published, this will include your full peer review and any attached files.

Reviewer #1: No

Reviewer #2: No

Reviewer #3: No

Reviewer #4: No

---

## [Author Response · Author response to Decision Letter 0]

10 Jul 2020

Reviewer #1: 

The authors compared On-pump vs. off Pump CABG in octgenarians of the Polish cardiac surgery registry. They found that on-Pump was associated with more perioperative complications and mortality but similar long term outcomes.

Comments

This is a well written manuscript, which is easy to comprehend. It addresses a classic but still relevant topic in cardiac surgery, i.e. the comparison of off-pump vs. on-pump bypass surgery. The data are valuable and important. However, the issue of conversions from off-pump to on-pump is not addressed. if this major confounder can be excluded, there remain only minor concerns.

Specific comments

1. One of the big problems in the comparison of Off-Pump to On Pump is the method of data comparison. In randomized trials with an intention to treat analysis, differences are often not visible, while in retrospective analyses, conversnions from Off- to On-Pump may affect the results. It is not clear in this dataset, how this confounder was handeled. Does the database record conversions? Were they excluded? How many were there? Is the higher mortality in th on-Pump group possibly due to those converted patients ? This needs to be clarified and discussed in detail.

Answer:

There is a field for conversions in the KROK database. Overall, 53 conversions were identified among 6,006 patients (0.9%), however the type of surgical procedure was marked differently in these patients. For patients with conversion, users of the KROK database chose either the finally performed surgical procedure (CABG) or the originally planned surgical procedure (OPCAB). To avoid any doubt with the grouping variable (OPCAB/CABG), all these patients now have been excluded. The number of patients was therefore reduced, and their total number was reduced from 6,006 to 5,953 patients. The percentage of in-hospital mortality in a subgroup of 53 patients with conversion was 32%. 

Action:

All patients with conversion have been excluded. All calculations have therefore been repeated, with appropriate changes in the text and all tables. An explanation of the reason for exclusion of patients with conversion has been added to the Methods section. 

2. The authors state, that it is more difficult to decide for surgery in the elderly today than it was previously. The argument is not convincing, when life expectancy is increaseing. Their figures even illustrate average life expectancy of this patient population in Poland which can easily be used to make recomendations for the only treatment of coronary artery disease with a prognostic impact demonstrated in prospective randomimzed trials.

Answer:

We are in agreement with this comment.

Action:

This sentence has been deleted from the text. 

Reviewer #2: 

The present study investigated the effect of off-pump versus on-pump isolated CABG surgery on short-term and long-term clinical outcome in the entire cohort of Polish octogenarians who underwent cardiac surgery between 2006 and 2017. It is concluded that the off-pump technique is associated with lower in-hospital mortality than the on-pump technique, whereas long-term mortality is similar between study groups.

General comments:

This is a large register analysis of Polish octogenarians. Since there is still a controversial debate regarding long-term clinical outcomes in patients undergoing off-pump or on-pump isolated CABG surgery, data are timely. However, there are issues that have to be addressed:

Statistical analysis of unadjusted data is subject to unexplained confounding. Therefore, comparison of unadjusted data regarding clinical outcomes are irrelevant. Of scientific interest regarding the effect of off-pump versus on-pump on clinical outcome are only the PS-matched data of the manuscript. Therefore, they should primarily present the PS-matched data both in the Results section as well as in the Discussion section. The manuscript should be revised accordingly. The entire study cohort can be used, however, to perform sensitivity analyses by presenting multivariable-adjusted data (see Puehler et al. Thorac Cardiovasc Surg. 2019 Feb 9. doi: 10.1055/s-0039-1677835. [Epub ahead of print].

Answer:

We fully agree, that only PS-matched data are relevant to draw conclusions from, thus our results and conclusions are based only on PS-matched data. The same is also true for the Discussion section, where we again, discuss only PS-matched data. 

We somewhat disagree however, that we do not need to provide unadjusted data at all. To the best of our knowledge, the authors should provide both unadjusted and PS-matched data, hence our provision of both dataset in the tables. 

Following a discussion with our statistician, we came to the conclusion that there was no need to perform sensitivity analysis by presenting multivariable-adjusted data. 

Action:

None

In the Methods section, they should clearly define primary and secondary endpoints. This is necessary because otherwise statistical methods are needed to consider the problem of multiple testing.

Answer:

In the Methods section of our study we stated the following: „The primary outcomes of this study were the in-hospital mortality rate and the incidence of perioperative complications. The secondary outcome was all-cause mortality in a long-term follow-up period.” Therefore, both the primary and secondary endpoints were clearly defined in the Methods section. The only difference was in using the word “outcome” instead of “endpoint”. We are happy to change this as per the reviewer’s suggestion. 

Action:

None

The method of PS-matching should be clearly described in the Statistics section. Moreover, the baseline characteristics used for PS-matching should be listed. Usually, standardized differences rather than p-values are used to compare baseline parameters between the two groups of the entire study cohort and the PS-matched groups. They should revise Table 1 accordingly.

Answer:

Patients for comparison were matched to achieve the similar pre-operative status. The data was matched with the Greedy data matching algorithm using Mahalanobis distance within propensity score calipers. Each caliper radius was set to 0.2*Sigma. Propensity score was calculated using logistic regression. We used all variables from Table 1. 

To assess the covariate balance, z-difference coefficients were calculated for each variable before and after matching. The mean value before and after the match was 0.28 and -0.30, respectively, and the variances were 79.31 and 0.32, respectively. Attached are detailed data about PSM.

We added the appropriate description in the Methods section and therefore, the method of PS-matching is clearly described.

Additionally, we fully agree with the Reviewer that standardized differences should be also used to compare baseline parameters between the two groups of the entire study cohort and the PS-matched groups. We have therefore expanded and revised Table 1 accordingly.

Action: 

The method of PS-matching has been clearly described in the Methods section. To attain this goal, we added the appropriate description in the Methods section.

We have also expanded Table 1 (adding two additional columns) to present standardized differences to compare baseline parameters between the two groups of the entire study cohort and the PS-matched groups.

Clinical outcomes may be influenced by year of surgery, surgeon’s experience, and number of diseased vessels. It is important to consider these parameters in the PS-matched cohort. If some of these data are not available, this should be mentioned as limitation in the Discussion section.

Answer:

We fully agree that clinical outcomes may be influenced by the year of surgery. However, dividing year of surgery into 12 separate variables (describing each consecutive year) does not seem like a good approach. Instead, we divided the analyzed time period into three relatively equal time periods (2006-2009, 2010-2013, 2014-2017), adding this information to a newly created part of table 1 (entitled “year of surgery’) and adding this variable to the PS-matching.

With regards to your point about surgeon’s experience, there is unfortunately nothing we can do to improve the quality of our calculations. Part of the limitations of the KROK database is that it doesn’t provide information about the surgical experience of individual Polish cardiac surgeons. Whilst we agree that such a parameter would be very useful, we regret that it is something we cannot obtain.

Number of diseased vessels does not exist in a KROK database as a separate variable, however, we have information available on the number of grafts. Therefore, we divided the amount of grafts performed per patient into three categories: “1 graft”, “2 grafts” and “3 or more grafts”. Three additional rows have been therefore added to “procedure related variables” in the table and these variables were also added to the PS-matching.

Action:

We divided the analyzed time period into three relatively equal time periods (2006-2009, 2010-2013, 2014-2017), adding this information to a newly created part of table 1 (entitled “year of surgery”) and adding this variable to the PS-matching. Also, we divided the amount of grafts performed per patient into three categories: “1 graft”, “2 grafts”, “3 or more grafts”. Additional rows have been therefore added to “procedure related variables” in table 1 and these variables were also added to the PS-matching.

Several perioperative data, which may substantially influence long-term clinical outcome, are missing and should be presented by study group. These parameters include the revascularization ratio, the percentage of complete arterial revascularizations, and the ratio of posterior to inferior anastomoses.

Answer:

Unfortunately, variables such as revascularization ratio and the ratio of posterior to inferior anasomoses are not available in the KROK Registry. We were able however, to identify, which patients underwent a complete arterial revascularization. Therefore, this variable was also added to table 1 (as another „procedure-related variable”) and was included in a PS-matching.

Action: 

Variable “complete arterial revascularization” was also added to table 1 (among “procedure-related variables”) and was also included in a PS-matching.

Although not restricted to octogenarians, they should refer to two PS-matched large studies: One study supports the early results of the present study regarding beneficial effects in the off-pump versus the on-pump group on clinical outcome (Hulde et al. Interact Cardiovasc Thorac Surg. 2020;30:538-540. The other large study, a US register analysis, reported higher mortality over 10-years of follow-up in the off-pump versus on-pump group (Chikwe et al. J Am Coll Cardiol. 2018;72:1478-1486.). Potential causes of the inconsistent results between the present data and the US data should be discussed more deeply.

Answer:

It has been done, according to the Reviewers’ suggestion.

Action:

Two consecutive paragraphs in the Discussion section have been extended to include this comment and the above mentioned two references have been added to the list of references. Additional sentences are added at the end of each paragraph.These paragraphs are now as follows:

“Our findings suggest that octogenarians constitute a high-risk population and the off-pump procedure appears superior among patients scheduled for isolated coronary artery bypass surgery for these patients in terms of in-hospital results. Utilization of the on-pump technique resulted in significantly higher in-hospital mortality (8.4% vs 4.7%, p<0.001) and similar all-cause mortality in the long-term follow-up period in survivors (p=0.362). The increased in-hospital mortality was likely due to the higher incidence of postoperative complications (mainly reoperation due to bleeding) in the on-pump group. Similar conclusions were drawn by Hulde et al. who observed that the duration of mechanical ventilation duration, intensive care unit stay, the risk of stroke, in-hospital mortality and 30-day mortality were significantly lower in the off-pump group than in the on-pump group (Hulde et al. Interact Cardiovasc Thorac Surg. 2020;30:538-540).

Comparative findings reported in the literature are contradictory. One recent retrospective study, analyzing 134.117 discharge records from 797 US hospitals, found that in-hospital mortality did not differ between octogenarians who underwent CABG and OPCAB (5.5% vs. 5.2%, p=0.3) [14]. At the same time, a recent systematic review based on 16 retrospective studies (performed in 27.623 octogenarians overall) found that the OPCAB technique was associated with significantly lower in-hospital mortality [9]. The other large study, a US register analysis by Chikwe et al., reported higher mortality over 10-years of follow-up in the off-pump versus on-pump group (Chikwe et al. J Am Coll Cardiol. 2018;72:1478-1486.)”.

Specific comments

A flowchart of included and excluded patients should be presented.

Answer:

It has been done.

Action:

A flowchart has been added in a form of Figure 1. The numbers of the remaining figures have been changed accordingly.

Completeness of follow-data should be calculated and presented in the manuscript (see Wu et al. Ann Thorac Surg 2008;85:1155-7).

Answer:

Completness of follow-up data have been calculated according to the method described by Wu et. al. We described the method of this assessment in the Methods section (adding an additional reference) and presented the results on the beginning of the Results section.

Additionally, we have noticed that we have specified a wrong closing date regarding the follow-up assessment. It has been now corrected in the text of the Methods section.

Action:

In the Methods section we added the following sentence: 

„Completness of follow-up data was calculated according to the method described by Wu et. al. [appropriate reference number will be cited here]”. 

In the Results section we added the following sentence: 

„Completness of follow-up data according to Clark’s C-index was 78.6% and the modified C*-index was 88.8%.”

Also, in the Methods section we have changed the following sentence:

“Assessment of long-term follow-up data included analysis of all-cause mortality. The National Health Fund death database was searched for all patients included in this study from the date of their procedure until 31st of March 2016”.

to:

“Assessment of long-term follow-up data included analysis of all-cause mortality. The National Health Fund death database was searched for all patients included in this study from the date of their procedure until 30the of September 2017”.

Change the titles of figures 3a+3b into ‘Probability of Survival’.

Answer:

It has been done.

Action:

Titles of Figures 3a and 3b have been changed in a figure legend and above the figures, according to the Reviewers’ suggestion.

Since they used the term ‘OPCAB’, they should also use the term ‘ONCAB’ instead of ‘CABG’ throughout the manuscript.

Answer:

We used the term “CABG” for on-pump coronary artery surgery, because this term is used by the Committee of the European Association for Cardio-Thoracic Surgery( EACTS) in the most recent Clinical Guidelines on Myocardial Revascularization from 2018. In these guidelines, the term CABG (not ONCAB) is used. We may change it, but we would be grateful if you could consider our point of view.

Action: 

None.

Reviewer #3: 

This is a retrospective review by Knapik and colleagues of isolated on- and off-pump CABG in octogenarians in Poland as obtained from the Polish National Registry of Cardiac Surgical Procedures (KROK). The primary aims of the study were to evaluate 1. perioperative (in -hospital mortality and complications) and 2. long term outcomes (all cause mortality).

Major Comments:

1. Why use the date of discharge as the starting point for survival; this is a somewhat odd choice? Isn't the day of operation more standard and more representative of survival from an operation? Were perioperative in-hospital deaths eliminated from the survival curves by using the day of discharge as the starting point? what is a patient was discharged 3 months after the operation, the day after discharge is survival day #1?

Answer:

Using the date of discharge as a starting point was intentional but we fully acknowledge the Reviewer’s point, especially with regard to the following comment “Perioperative in-hospital deaths were eliminated from the survival curves by using the day of discharge as the starting point”. 

We feel that using the day of operation as a starting point might create a lot of bias in this particular group. In case of significant in-hospital mortality and relatively frequent postoperative complications (among our octogenarians undergoing CABG), this would create an influence of in-hospital mortality on long-term results. The survival curves would in this case, separate at the beginning and then remain parallel during the whole postoperative course (even when in fact there was no difference in the long-term results). There are examples in the medical literature, where such an approach led to drawing wrong conclusions (see Masyuk et al. Intensive Care Med. 2019;45:55 and our commentary in Knapik et al. Intensive Care Med. 2019 Aug;45:1172.)

If we adopted the strategy proposed by the Reviewer, we would probably have significantly worse long-term results in the CABG group, because in-hospital mortality in this group was much higher and the curve would start on the day of surgery. We are therefore convinced that choosing the day of operation as a starting point should not be considered as a gold standard in all surgical procedures. Furthermore, we would like to mention that the same solution (taking the date of discharge as a starting point) has been previously successfully used – for the same reason – in our paper published in the Interactive Cardiovascular and Thoracic Surgery in 2019, when we analyzed reoperations due to postoperative bleeding on a basis of data from the KROK Registry (Knapik et al. Interact Cardiovasc Thorac Surg. 2019 Apr 9;ivz089. doi: 10.1093/icvts/ivz089). 

Action:

None

2. A statement that informed consent was waived by whatever appropriate review board should be included in the paper (if not obtained then rationale for why not should be included but hard to justify)

Answer: 

Informed consent was waived by our Ethical Committee. 

Action:

The following sentence has been added to the Methods section: “Due to the retrospective and anonymous nature of the study, Ethical Committee of the Medical University of Silesia in Katowice waived the need for consent of the patients to participate in the study.”

3. On line 165, you report the mean age was only 82 which means that the vast majority of patients with <85 years old. In Table 1, you report only 39 patients were over 90; can you include additional information about the number of patients over 85 and less than 90? This will help surgeons when applying data to their practices about what "octogenarians" you are operating on.

Answer:

An additional age category of patient between 86 – 90 years has been additionally specified as a separate variable.

Action:

Table 1 has been revised accordingly. This variable has been also added to PS-matching. 

4. Can you provide any information as to surgeon performance of both procedures? For instance, did surgeons performing OPCAB represent only a small percentage of total surgeons performing most of the operations or were OPCAB's performed by a wide variety of surgeons in a small percentage of patients? It would be important to know if the outcomes could be generalized to essentially any surgeon even if less experienced. Near the end of the discussion, you mention that surgeons' rationale for procedure selection is unknown and also there were centers where OPCAB was widely used (and presumably some where it was rarely used) Can you provide any information as to variation by institution? by surgeon?

Answer:

Based on the data from the KROK registry, we are unable to determine the degree of experience of cardiac surgeons in performing various types of coronary procedures. One could probably get information on what percentage of off-pump coronary surgery was performed in the individual centers and, accordingly, carry out appropriate PS-matching, but this might be also prone to serious bias. For example, there might be centers that used to carry out only few off-pump procedures in the past, where the surgical team was suddenly joined by one or two surgeons skilled in off-pump surgery. The same mechanism may also work the other way. The results may rapidly change in both cases and thus, such analysis does not seem logical.

Action:

None

5. Although a Kaplan Meier curve can be used to depict the time on ventilator following surgery, the inclusion of 2 figures just to show this before and after propensity matching seems excessive; please remove and instead pick one or two additional individual time points to demonstrate the difference (you talk about the 24 hr time point already in the text). Also, please comment on the "significance" of the difference (13.0% versus 8.2% after 24 hrs); was there higher ventilator associated pneumonias in the patients remaining ventilated. Was there still a significant difference after 48 hrs? What about after 3 or more days?

Answer:

It is true that having two curves showing ventilation time (before and after PS-matching) is indeed excessive. Therefore, one curve (showing ventilation times before PS-matching) has been removed. 

It was also suggested that we pick one or two individual time points to demonstrate the difference. Our issue with that is that in our opinion, analysis of the duration of postoperative ventilation makes more sense in the first 24 hours after surgery, for example 12 hours after the end of the procedure. This moment however, is clearly seen on the existing curves. 

In our view, a higher percentage of patients still being ventilated in the following days would have a relationship primarily with postoperative complications. Also, patients who die in the postoperative period may have had relatively short ventilation times. Therefore, we decided not to analyze a more distant postoperative period in terms of postoperative ventilation

Action:

One curve (showing ventilation times before PS-matching) was removed.

6. The statement on lines 353-355 "This is a situation when locally obtained, retrospective data should be treated with the same attention as the results of a perfectly designed prospective, randomized trials." is not consistent with known biases that occur with patient selection in retrospective trials which cannot be consistently corrected with "propensity matching" This comment should be removed as "perfectly planned prospective randomized trials remain the gold standard.

Answer:

It is true that perfectly planned prospective randomized trials remain a gold standard. We have therefore decided to change this sentence to: “Such strong scientific evidence should lead to careful evaluation of the optimal method of revascularization in this group of patients in Poland. This is also a situation when locally obtained, retrospective data should be taken seriously.”

Action:

The following piece of text has been modified in the Discussion section: „Such strong scientific evidence should lead to careful evaluation of the optimal method of revascularization in this group of patients in Poland. This is also a situation when locally obtained, retrospective data are of great practical importance.”

7. Your conclusion paragraph states a little too strongly that OPCAB should be the method of choice in octogenarians and this statement and the data may not apply to certain subsets of patient where on pump CABG would be a better choice. This statement should be softened to something like "suggest that the OPCAB technique should be considered as perfectly acceptable in octogenarians when performed by surgeons experienced in the technique."

Answer:

We fully agree that our conclusion strongly states that OPCAB should be the method of choice in octogenarians. Therefore, we have decided to change the conclusion according to the Reviewer’s suggestion.

Action: 

At the end of the Discussion section, the following sentence replaced the previous statement: “Based on the results of this study, we suggest that the OPCAB technique should be considered as perfectly acceptable in octogenarians when performed by surgeons experienced in the technique”.

Also, in the conclusion of the abstract, the sentence has been changed to: “On the basis of our findings we suggest that off pump technique should be considered as perfectly acceptable in octogenarians.”

Minor Comments:

1. Table 2 title: please correct spelling of "patiets" to patients

Answer:

We agree with this comment.

Action:

It has been corrected.

2. On lines 223-24 the statement ""the incidence of the some postoperative complications was still higher in the on-pump group." does not make sense; please correct.

Answer:

We agree with this comment.

Action:

It has been corrected to: „Following propensity matching, the incidence of postoperative respiratory complications and reoperation due to bleeding was higher in the on-pump group”.

Reviewer #4: PONE-D-20-11113: statistical review

SUMMARY. This is a retrospective study that investigates in-hospital and long-term mortality among octogenarians undergoing off-pump and on-pump coronary artery bypass surgery. The core statistical analysis is based on the comparison of survival curves under off-pump or on-pomp treatments. Although the results seem sound and the material is well organized, the paper lacks details (major issue 1) and the data are not provided (major issue 2). These issues complicate not only the interpretation of the results but also their reproducibility. I also list a couple of specific points that should be addressed.

MAJOR ISSUES

1. Propensity score matching. The paper does not provide enough details about the methods used, making it impossible to reproduce the results. Please provide these details and especially clarify (1) whether you used a logistic regression model to estimate the propensity score (in this case, please provide the estimates, perhaps as supplementary material) and (2) the covariates that have been used (did you consider any variable selection method?).

Answer:

The issue regarding the details regarding propensity score matching has been already raised by Reviewer 2. We therefore added the appropriate sentences to the Methods section. The method of PS-matching is now described in detail in this chapter. 

Action: 

The following sentence has been added to the Methods section: 

“Data were matched with the Greedy data matching procedure using Mahalanobis distance within propensity score calipers. Caliper radius were set to 0.2*Sigma. Propensity score was calculated using logistic regression. We used all variables from table 1. To assess the covariate balance, z-difference coefficients were calculated for each variable before and after matching. The mean value before and after the match was 0.28 and -0.30, respectively, and the variance was 79.31 and 0.32.”

Details regarding logistic regression model and covariates have been attached in the supplementary file 1.

2. Data availability. Although the authors declare that the data are available without restrictions and that they are within the manuscript and its Supporting Information files, data are not attached. Data should be provided as a supplementary information file along with the metadata needed to process the file.

Answer:

The Polish National Registry of Cardiac Surgical Operations (KROK Registry) operates under the supervision of the Polish Ministry of Health and the scope of data subject to public disclosure is strictly defined. Data allowed to be made publicly available have been now attached together with the description. These data enable repetition of our follow-up analysis.

Action:

Data allowed to be publicly available have been now attached together with the description as Supplementary file 2. These data enable repetition of our follow-up analysis.

SPECIFIC POINTS

1. Line 152 “Multivariate analysis of long-term results was performed with the use of the Cox-proportional hazard model.” I can’t see the results of this analysis in the paper. Please clarify.

Answer:

Multivariate analysis of long-term results was not performed with the use of Cox-proportional hazard model. This sentence was found in the text of the study by mistake. We would like to apologize for that.

Action:

The sentence „Multivariate analysis of long-term results was performed with the use of the Cox-proportional hazard model” was deleted from the text of the Methods section.

2. Line 150 “These data were then analyzed using the Kaplan-Meier method with log-rank testing.”. What kind of log-rank test? While the standard log-rank test is frequently used for testing the equality of survival curves in propensity score matched samples, such an approach is inappropriate, because it requires that the samples be independent of one another. Instead, the stratified log-rank test can be used to compare the equality of the survival curves in matched samples.

Answer:

Log-rank test which has been used is the stratified log-rank test. 

Action:

The information regarding the type of log-rank test was added to the appropriate sentence in the Methods section.

---

## [Decision Letter · Decision Letter 1]

29 Jul 2020

PONE-D-20-11113R1

Off-pump versus on-pump coronary artery surgery in octogenarians (from the KROK Registry)

PLOS ONE

Dear Dr. Knapik,

Thank you for submitting your manuscript to PLOS ONE. After careful consideration, we feel that it has merit but does not fully meet PLOS ONE’s publication criteria as it currently stands. Therefore, we invite you to submit a revised version of the manuscript that addresses the points raised during the review process.

We look forward to receiving your revised manuscript.

Kind regards,

Mariusz Kowalewski

Academic Editor

PLOS ONE

Additional Editor Comments (if provided):

We are interested in publishing your manuscript; however, one reviewer continues to have significant concerns with your revised manuscript. Before the paper can be accepted in its final form we would like your comments to the critiques. Accordingly, we invite you to respond to all the reviewers' comments and recommendations. A decision on acceptability of your manuscript will be made only after the revised version has been reevaluated.

In particular, the Reviewer is concerned about exclusion of conversion cases and censoring the time-to-event analysis by the date of discharge. While removing in-hospital mortality from the analysis may better help understand the postoperative sequelae in these patients unobscured by features inherent to the in-hospital course, the principal analysis should include both in-hospital and long-term; i therefore suggest to keep censored analysis to the supplement only but cite the results in the respective paragraph, similarly for the conversion cases, please report the mortality rates for both per-protocol and ITT scenarios.

Reviewers' comments:

Reviewer's Responses to Questions

**Comments to the Author**

1. If the authors have adequately addressed your comments raised in a previous round of review and you feel that this manuscript is now acceptable for publication, you may indicate that here to bypass the “Comments to the Author” section, enter your conflict of interest statement in the “Confidential to Editor” section, and submit your "Accept" recommendation.

Reviewer #1: All comments have been addressed

Reviewer #3: (No Response)

Reviewer #4: All comments have been addressed

2. Is the manuscript technically sound, and do the data support the conclusions?

Reviewer #1: Yes

Reviewer #3: No

Reviewer #4: (No Response)

3. Has the statistical analysis been performed appropriately and rigorously? 

Reviewer #1: Yes

Reviewer #3: Yes

Reviewer #4: (No Response)

4. Have the authors made all data underlying the findings in their manuscript fully available?

Reviewer #1: Yes

Reviewer #3: Yes

Reviewer #4: (No Response)

5. Is the manuscript presented in an intelligible fashion and written in standard English?

Reviewer #1: Yes

Reviewer #3: Yes

Reviewer #4: (No Response)

6. Review Comments to the Author

Reviewer #1: The authors adequately addressed all of the concerns. Nevertheless, there remains a certain risk of bias due to multiple confounders. Therefore, the authors should consider to tune down their conclusion.

Reviewer #3: 1. The elimination of the conversion patients (which should be almost all OPCAB converted to CABG since many fewer might be converted to OPCAB from CABG due to porcelain aorta, etc. I do not believe that these patients should be excluded because they have a very high mortality rate which therefore biases the results if these were intended to be OPCAB patients!

2. The authors explanation for the bias introduced by using the date of discharge as the start of survival is unsupportable. The elimination of mortality by this method cannot be avoided by surgeons and therefore to say that "if you make it to discharge" that OPCAB is more likely to survive (but the opposite is true if you take into account the mortality of conversions and perioperative deaths) is erroneous and not acceptable.

Reviewer #4: (No Response)

7. PLOS authors have the option to publish the peer review history of their article (what does this mean?). If published, this will include your full peer review and any attached files.

Reviewer #1: No

Reviewer #3: No

Reviewer #4: No

---

## [Author Response · Author response to Decision Letter 1]

24 Aug 2020

Additional Editor Comments (if provided):

We are interested in publishing your manuscript; however, one reviewer continues to have significant concerns with your revised manuscript. Before the paper can be accepted in its final form we would like your comments to the critiques. Accordingly, we invite you to respond to all the reviewers' comments and recommendations. A decision on acceptability of your manuscript will be made only after the revised version has been reevaluated.

In particular, the Reviewer is concerned about exclusion of conversion cases and censoring the time-to-event analysis by the date of discharge. While removing in-hospital mortality from the analysis may better help understand the postoperative sequelae in these patients unobscured by features inherent to the in-hospital course, the principal analysis should include both in-hospital and long-term; i therefore suggest to keep censored analysis to the supplement only but cite the results in the respective paragraph, similarly for the conversion cases, please report the mortality rates for both per-protocol and ITT scenarios.

Answer:

As previously explained, 53 conversions were identified among 6,006 patients (0.9%), however the type of surgical procedure was marked differently in these patients. For patients with conversion, users of the KROK database chose either the finally performed surgical procedure (CABG) or the originally planned surgical procedure (OPCAB). Therefore, to avoid any doubt with the grouping variable (OPCAB/CABG), we proposed to exclude these patients. The number of patients was therefore reduced, and their total number was reduced from 6,006 to 5,953 patients.

We understand however, that the Reviewer might have been concerned about exclusion of conversion cases, as the in-hospital mortality in this subgroup of 53 patients was as high as 32%. Therefore, we decided to follow the intention-to-treat approach and all patients with conversion were allocated in the OPCAB group. In the Results section however, we decided to present mortality rates for both per-protocol and ITT scenario. 

Additionally, the Reviewer was also concerned about censoring the time-to-event analysis by the date of discharge, and indicated that the principal analysis should include both in-hospital and long-term results.

While we generally agree with this view, we feel that using the day of operation as a starting point might create a lot of bias in this particular situation. The reasons for that have been already explained. In case of significant in-hospital mortality and relatively frequent postoperative complications (among our octogenarians undergoing CABG), this would create an influence of in-hospital mortality on long-term results. The survival curves would in this case, separate at the beginning and then remain parallel during the whole postoperative course (even when in fact there was no difference in the long-term results). It turned out, that we were right – when the starting point of our analysis was changed, the survival curves following propensity scoring starting at the day of operation indicated that long-term survival was significantly better (p=0.013) in the OPCAB group (despite the fact that all conversions are being now analyzed in the OPCAB group).

We understand however, that the Reviewer might have been concerned with our previous approach. Therefore, we performed the principal analysis starting at the day of the operation and took Editors’ comment to keep censored analysis to the supplement only. 

Action: 

All patients with conversion were allocated in the OPCAB group. As a consequence, all results had to be recalculated and new data are now presented in all tables and figures, and appropriate changes were also made in the whole text of the manuscript. Additionally, in the Results section, we presented mortality rates for both per-protocol and ITT scenario, as required.

Also, we performed the principal analysis starting at the day of the operation. Additionally, we presented our censored analysis to the supplement only (as Supplementary file 3), presenting the results of this additional analysis in the Results section. On a basis of these calculations, we had to change one of the results of our study (regarding a long-term follow up). 

Reviewers' comments:

Reviewer's Responses to Questions

Comments to the Author

1. If the authors have adequately addressed your comments raised in a previous round of review and you feel that this manuscript is now acceptable for publication, you may indicate that here to bypass the “Comments to the Author” section, enter your conflict of interest statement in the “Confidential to Editor” section, and submit your "Accept" recommendation.

Reviewer #1: All comments have been addressed

Reviewer #3: (No Response)

Reviewer #4: All comments have been addressed

2. Is the manuscript technically sound, and do the data support the conclusions?

Reviewer #1: Yes

Reviewer #3: No

Reviewer #4: (No Response)

3. Has the statistical analysis been performed appropriately and rigorously? 

Reviewer #1: Yes

Reviewer #3: Yes

Reviewer #4: (No Response)

4. Have the authors made all data underlying the findings in their manuscript fully available?

Reviewer #1: Yes

Reviewer #3: Yes

Reviewer #4: (No Response)

5. Is the manuscript presented in an intelligible fashion and written in standard English?

Reviewer #1: Yes

Reviewer #3: Yes

Reviewer #4: (No Response)

6. Review Comments to the Author.

Reviewer #1: 

The authors adequately addressed all of the concerns. Nevertheless, there remains a certain risk of bias due to multiple confounders. Therefore, the authors should consider to tune down their conclusion.

Answer:

This issue has been already raised in a previous round by another Reviewer who stated: „Your conclusion paragraph states a little too strongly that OPCAB should be the method of choice in octogenarians and this statement and the data may not apply to certain subsets of patient where on pump CABG would be a better choice. This statement should be softened…”

On a basis of this comment, we have decided to change the conclusion according to the Reviewer’s suggestion. Therefore, the conclusion has been already tuned down during a previous review round.

Action:

At the end of the Discussion section, the following sentence replaced the previous statement: “Based on the results of this study, we suggest that the OPCAB technique should be considered as perfectly acceptable in octogenarians when performed by surgeons experienced in the technique”.

Also, in the conclusion of the abstract, the sentence has been changed to: “On the basis of our findings we suggest that off pump technique should be considered as perfectly acceptable in octogenarians.”

These changes have been already done during a previous review round.

Reviewer #3: 

1. The elimination of the conversion patients (which should be almost all OPCAB converted to CABG since many fewer might be converted to OPCAB from CABG due to porcelain aorta, etc. I do not believe that these patients should be excluded because they have a very high mortality rate which therefore biases the results if these were intended to be OPCAB patients!

Answer:

As explained during a previous review round, 53 conversions were identified among 6,006 patients (0.9%), however the type of surgical procedure was marked differently in these patients. For patients with conversion, users of the KROK database chose either the finally performed surgical procedure (CABG) or the originally planned surgical procedure (OPCAB). Therefore, to avoid any doubt with the grouping variable (OPCAB/CABG), we initially proposed to exclude these patients. The number of patients was therefore reduced, and their total number was reduced from 6,006 to 5,953 patients in a previous version of the manuscript.

We understand however, that exclusion of conversion cases might not be acceptable, particularly when the in-hospital mortality in this subgroup of 53 patients was as high as 32%. Therefore, we decided to follow the intention-to-treat approach and all patients with conversion were allocated in the OPCAB group. In the Results section however, we decided to present mortality rates for both per-protocol and ITT scenario, as required. 

Action: 

All patients with conversion were allocated in the OPCAB group. As a consequence, all results had to be recalculated and new data are now presented in all tables and figures, and appropriate changes were also made in the whole text of the manuscript. Additionally, in the Results section, we presented mortality rates for both per-protocol and ITT scenario (as proposed by the Editor).

In the Results section the following sentence was therefore added: “Conversions from OPCAB do CABG were analyzed in the OPCAB group”.

2. The authors explanation for the bias introduced by using the date of discharge as the start of survival is unsupportable. The elimination of mortality by this method cannot be avoided by surgeons and therefore to say that "if you make it to discharge" that OPCAB is more likely to survive (but the opposite is true if you take into account the mortality of conversions and perioperative deaths) is erroneous and not acceptable.

Answer:

We understand, that the Reviewer might have been concerned with our approach, when the survival curves start at the date of discharge. Therefore, we now performed the principal analysis starting at the day of the operation and took Editors’ comment to keep censored analysis to the supplement only, presenting the results of this additional analysis in the respective paragraph. 

Action: 

We recalculated our data and performed the principal analysis starting at the day of the operation. Additionally, we presented our censored analysis to the supplement only (adding Supplementary file 3), presenting the results of this additional analysis in the Results section. Changes were also done in the whole text of the manuscript, as appropriate.

When the starting point of our analysis was changed, the survival curves following propensity scoring starting at the day of operation indicated that long-term survival was significantly better (p=0.013) in the OPCAB group (despite the fact that all conversions are being now analyzed in the OPCAB group). Therefore, we had to do some minor changes in the text of the manuscript (in the abstract, and in the Results and Discussion section).. 

Reviewer #4: (No Response)

7. PLOS authors have the option to publish the peer review history of their article (what does this mean?). If published, this will include your full peer review and any attached files.

Do you want your identity to be public for this peer review? For information about this choice, including consent withdrawal, please see our Privacy Policy.

Reviewer #1: No

Reviewer #3: No

Reviewer #4: No

---

## [Editor Report · Decision Letter 2]

26 Aug 2020

Off-pump versus on-pump coronary artery surgery in octogenarians (from the KROK Registry)

PONE-D-20-11113R2

Dear Dr. Knapik,

We’re pleased to inform you that your manuscript has been judged scientifically suitable for publication and will be formally accepted for publication once it meets all outstanding technical requirements.

Kind regards,

Mariusz Kowalewski

Academic Editor

PLOS ONE

Additional Editor Comments (optional):

No other comments from academic editor
---

## [Editor Report · Acceptance letter]

28 Aug 2020

PONE-D-20-11113R2 

Off-pump versus on-pump coronary artery surgery in octogenarians (from the KROK Registry) 

Dear Dr. Knapik:

I'm pleased to inform you that your manuscript has been deemed suitable for publication in PLOS ONE. Congratulations! Your manuscript is now with our production department. 

Kind regards, 

on behalf of

Dr. Mariusz Kowalewski 

Academic Editor

PLOS ONE